# Pre-Referral Primary Care Blood Tests and Symptom Presentation before Cancer Diagnosis: National Cancer Diagnosis Audit Data

**DOI:** 10.3390/cancers15143587

**Published:** 2023-07-12

**Authors:** Ben M. Cranfield, Gary A. Abel, Ruth Swann, Sarah F. Moore, Sean McPhail, Greg P. Rubin, Georgios Lyratzopoulos

**Affiliations:** 1Epidemiology of Cancer Healthcare and Outcomes (ECHO) Research Group, Department of Behavioural Science and Health, University College London, 1-19 Torrington Place, London WC1E 6BT, UK; y.lyratzopoulos@ucl.ac.uk; 2University of Exeter Medical School, St Luke’s Campus, Exeter EX1 2HZ, UK; 3National Disease Registration Service, NHS England, Leeds LS1 4AP, UK; 4Cancer Research UK, London E20 1JQ, UK; 5Population Health Sciences Institute, Newcastle University, Newcastle upon Tyne NE1 4LP, UK

**Keywords:** blood tests, cancer, symptoms, national cancer diagnosis audit

## Abstract

**Simple Summary:**

Blood tests can support decisions by GPs about referring patients who present with symptoms of possible cancer for specialist assessment. This study analysed data on the use of blood tests in primary care in patients subsequently diagnosed with cancer to understand how often and when blood tests were used. We found that the use of generic blood tests (including full blood count, urea and electrolyte, liver function, and inflammatory marker tests) varied widely between patients presenting with different symptoms, with greater use in patients presenting with certain nonspecific symptoms (e.g., fatigue or loss of weight) and least frequently in those presenting with certain red-flag symptoms (e.g., breast or skin symptoms). Blood tests with greater specificity to certain organs/pathologies (including serum protein electrophoresis, ferritin, bone profile, and amylase tests) followed a similar use pattern regarding symptom specificity but at a lower use frequency. Commonly used cancer biomarkers were used in varying proportions depending on whether the presenting symptom could be related to prostate or ovarian cancer (for example, 88% of men presenting with lower urinary tract symptoms had presumed PSA measurement). The findings benchmark how often blood tests are used in certain clinical scenarios and identify opportunities for greater use in patients with symptoms of low (<3%) positive predictive value for cancer.

**Abstract:**

Background: Blood tests can support the diagnostic process in primary care. Understanding how symptomatic presentations are associated with blood test use in patients subsequently diagnosed with cancer can help to benchmark current practices and guide interventions. Methods: English National Cancer Diagnosis Audit data on 39,751 patients with incident cancer in 2018 were analysed. The frequency of four generic (full blood count, urea and electrolytes, liver function tests, and inflammatory markers) and five organ-specific (cancer biomarkers (PSA or CA125), serum protein electrophoresis, ferritin, bone profile, and amylase) blood tests was described for a total of 83 presenting symptoms. The adjusted analysis explored variation in blood test use by the symptom-positive predictive value (PPV) group. Results: There was a large variation in generic blood test use by presenting symptoms, being higher in patients subsequently diagnosed with cancer who presented with nonspecific symptoms (e.g., fatigue 81% or loss of appetite 79%), and lower in those who presented with alarm symptoms (e.g., breast lump 3% or skin lesion 1%). Serum protein electrophoresis (reflecting suspicion of multiple myeloma) was most frequently used in cancer patients who presented with back pain (18%), and amylase measurement (reflecting suspicion of pancreatic cancer) was used in those who presented with upper abdominal pain (14%). Prostate-specific antigen (PSA) use was greatest in men with cancer who presented with lower urinary tract symptoms (88%), and CA125 in women with cancer who presented with abdominal distention (53%). Symptoms with PPV values between 2.00–2.99% were associated with greater test use (64%) compared with 52% and 51% in symptoms with PPVs in the 0.01–0.99 or 1.00–1.99% range and compared with 42% and 31% in symptoms with PPVs in either the 3.00–4.99 or ≥5% range (*p* < 0.001). Conclusions: Generic blood test use reflects the PPV of presenting symptoms, and the use of organ-specific tests is greater in patients with symptomatic presentations with known associations with certain cancer sites. There are opportunities for greater blood test use in patients presenting with symptoms that do not meet referral thresholds (i.e., <3% PPV for cancer) where information gain to support referral decisions is likely greatest. The findings benchmark blood test use in cancer patients, highlighting opportunities for increasing use.

## 1. Background

Most patients subsequently diagnosed with cancer first present with symptoms to their GP [1,2]. The nature of presenting symptoms is important for clinicians’ decisions on diagnostic management. For ‘alarm’ symptoms with relatively high predictive value for cancer, recommendations exist for urgent referrals for suspected cancer [3]. However, half of the patients with as-yet-undiagnosed cancer present with symptoms of lower specificity, for which clinical guideline recommendations about optimal management are lacking [4]. These patients typically experience longer intervals to diagnosis and are less likely to be referred to fast-track investigative pathways [5,6,7]. Evidence on the predictive value of blood tests for cancer supports their use in patients presenting with nonspecific symptoms [8,9,10,11,12,13,14,15]. Nonetheless, these potential benefits are contingent on the blood tests being ordered. Exploring variation in blood test use by presenting symptoms among patients subsequently diagnosed with cancer can provide a better understanding of the current clinical practice and determinants of blood test use. While certain predictors of primary care blood testing in cancer patients have been described [16], a detailed understanding of variation in their use by specific presenting symptoms is lacking. By generating evidence to address this gap, symptoms in which blood test use can be increased can be identified.

## 2. Methods

### 2.1. Study Design and Participants

Data were analysed from the (English) National Cancer Diagnosis Audit (NCDA) 2018. The nature of the data source, the characteristics of the study sample, and methodologies for sample definition and data collection have been described previously [16,17]. Data on the diagnostic process of 64,489 cancer cases diagnosed during 2018 were collected by participating GPs based on information in primary care records. Included patients were identified by the National Disease Registration Service responsible for cancer registration and were representative of the incident population of cancer patients in England. Participating general practices had comparable characteristics to nonparticipating practices though they were slightly larger [17].

The analysis sample included 39,751 non-screen-detected cancer patients aged 15 years or older, who first presented to general practice and for which there was complete information on investigation status (Derivation Sample: Figure 1).

### 2.2. Variables of Interest

The audit questionnaire collected information from patients’ medical records on whether blood tests were used in primary care pre-diagnosis using a relevant stem question (“Primary care led investigations that were ordered as part of the diagnostic assessment, and prior to referral, decided by the GP and in response to symptoms complained of, signs elicited, or abnormal test results”) with subsequent yes/no items for 4 generic blood tests (full blood count (FBC), urea and electrolytes (U&E), liver function tests (LFTs), and inflammatory markers (IM), and 5 organ-specific blood tests (cancer biomarkers, serum protein electrophoresis, ferritin, bone profile, and serum amylase). 

Information on presenting symptoms during patients’ initial consultation was collected from responses to drop-down menu items pertaining to 83 pre-specified symptoms. Of these, 37 presenting symptoms were recorded in at least 1% of the study population and were treated as separate categories; 46 were recorded in <1% of the study population and were grouped into an “all other symptoms” category, which accounted for 17% of all cases. 

In a supplementary analysis, symptoms were categorised into five groups by positive predictive value (PPV) for cancer, 0.01–0.99%, 1.00–1.99%, 2.00–2.99%, 3.00–4.99% and ≥5%, based on prior research [18] and clinical guidelines [3]. This analysis included a total of 42 symptoms (total n = 29,043, 73% of study population) for which there was previously published information to enable such classification—please see Appendix A. 

### 2.3. Analysis

We analysed the distribution of blood tests by presenting symptom categories. Analysis of cancer biomarkers was stratified by sex (i.e., assuming PSA testing in men and assumed CA125 testing in women). 

In the supplementary analysis, we described proportions of test use and used logistic regression to estimate relevant crude and adjusted odds ratios (ORs) use (excluding blood biomarkers) for symptoms in different PPV range groups (0.01–0.99%, 1.00–1.99%, 2.00–2.99%, 3.00–4.99% and ≥5%), age group (15–29, 30–49, 50–69, ≥70 years), sex (male and female) and index of multiple deprivation quintile group (based on income domain). Another model was further adjusted for cancer sites to explore the influence of cancer-specific factors on blood test use by the symptom PPV group. Joint Wald tests were used to assess overall variation by the variable category.

## 3. Results

### 3.1. Generic Blood Tests

Our analysis included 39,751 incident cancer cases. Across all patients, overall use of FBC, U&Es, LFTs, and IM tests was 39%, 37%, 31%, and 19%, respectively. There was a large variation in the use of these blood tests by presenting symptoms following a pattern whereby symptoms of lower specificity (e.g., fatigue, loss of appetite, and weight loss) were associated with the greatest use (Table 1a and Figure 2). In contrast, organ-specific symptoms, including those related to the skin and breast, were associated with the lowest frequency of blood test use (≤3%, Table 1a, Figure 2).

About one in five patients presenting with symptoms had IM tests (Table 1a). The use of IM tests followed a pattern of relative variation that was similar to that observed for FBC, U&Es, and LFTs (i.e., reflecting symptom specificity) but at a lower absolute frequency. 

### 3.2. Specific Blood Tests

Bone profile, ferritin, serum protein electrophoresis, and amylase tests were used less frequently than generic blood tests (Table 1b), the proportion of tested patients being 11% for bone profile and ferritin, 3% for serum protein electrophoresis, and 2% for amylase. 

For all these four tests, use followed a similar pattern to that observed for common blood tests, where less specific symptoms (including fatigue, loss of appetite, and weight loss) were generally associated with higher use, and vice versa. However, the absolute percentage of tested patients for these 3 nonspecific symptoms was much greater for bone profile (29%, 29%, and 26%, respectively) and ferritin (33%, 30%, and 27%, respectively) than serum protein electrophoresis (10%, 7%, and 7%, respectively) and amylase (4%, 7%, and 5%, respectively). 

Two tests, serum protein electrophoresis and bone profile, were used most commonly in patients with back pain (18% and 30%, respectively) or bone pain (16% and 32%), compared with percentages <10% in all patients presenting with all other symptoms. Amylase tests were more commonly used in patients presenting with upper abdominal pain (14%) compared with <9% of patients with all other symptoms.

### 3.3. Blood Biomarker Tests

Around one in three men who were subsequently diagnosed with cancer had a PSA test as part of their diagnostic process in primary care (see Table 2). PSA testing was chiefly concentrated in patients with urological symptoms, such as LUTS, dysuria, and UTI symptoms, and haematuria (35–88%). PSA use was also high among men presenting with back pain and bone pain (35% and 47%, respectively), and among men presenting asymptomatically, i.e., not-known—N/K symptoms (56%), or ‘not-applicable—N/A’ symptoms (63%).

Eight percent of women who were subsequently diagnosed with cancer had a CA125 test as part of their diagnostic process in primary care. CA125 testing was chiefly concentrated in patients with abdominal or urinary symptoms, including abdominal distension (53%), lower abdominal pain (32%), abdominal pain (30%) constipation (26%), LUTS (26%), and changes in bowel habit (22%). Several nonspecific symptoms, such as loss of appetite and weight loss, also had relatively high use (in around one in five women presenting with them). 

### 3.4. Supplementary Analysis: Generic and Less Common Blood Test Use by Symptom PPV Group 

A bimodal association was observed between increasing PPV range and test use, whereby use peaked in symptoms with PPV in the range of 2.00–2.99% and tailed off thereafter. Specifically, more than half (51–64%) of patients presenting with symptoms with PPVs ranging from 0.01–2.99% had a blood test, while the respective proportions were around two-fifths (42%) of those with PPVs between 3.00–4.99% and about one in three (31%) cases for symptoms with PPVs ≥5% (see Table 3). Multivariate analysis using the ≥5% PPV group as a reference provided concordant findings, with OR values of 2.40 (95% confidence interval [CI] = 2.21–2.60), 2.33 (CI = 2.19–2.48), and 3.89 (CI=3.61–4.18) for PPV range groups 0.01–0.99, 1.00–1.99, and 2.00–2.99%, respectively. In the 3.00–4.99% PPV range, the OR for use was 1.47 (1.34–1.62). Differences were attenuated and remained large and significant after adjustment for the cancer site to 1.57 (0.01–0.99 PPV; CI = 1.43–1.73), 1.96 (1.00–1.99 PPV; CI = 1.81–2.12), 2.79 (2.00–2.99 PPV; CI = 2.55–3.05), and 1.25 (3.00–4.99 PPV; CI = 1.11–1.40).

## 4. Discussion

Among patients subsequently diagnosed with cancer, blood test use as part of the diagnostic process in primary care is largely determined by the nature of presenting symptoms. The perceived organ specificity (or lack of organ specificity) of presenting symptoms regarding a possible cancer site seems to be the key driver of variation in the use of both generic and non-generic blood tests. Additionally, the affinity of non-generic tests to the presenting features of certain cancer sites has influenced their use.

Our analysis benefits from data obtained from a large and nationally representative sample of cancer patients [19]. Yet the precise timeframe between relevant symptom presentation and blood test use is not captured in the NCDA, although auditors were specifically requested to enter information on blood tests ordered before a referral decision. The availability of free-text information in the reviewed patient records may have improved the quality of symptom data capture compared with other sources of routine electronic health records data that predominantly rely on structured (coded) information, therefore improving the accuracy of their relationship with blood test use. The limitation of this study is that it is a case-only analysis; therefore, we are not able to decipher the frequency of use of the examined blood tests among patients who presented with the same symptoms but did not have underlying cancer. The findings add to earlier work that also described variation in blood test use by demographic characteristics, cancer site and presenting symptom category [16], by detailing observed variation by presenting symptom. 

Generally, blood testing was greatest after nonspecific presentations and lowest for organ-specific (‘alarm’) symptoms. For example, back pain (FBC use 60%) has a positive predictive value (PPV) of 0.1% for myeloma, for which reason additional information from blood test results is helpful, e.g., combined back pain and hypercalcaemia has a PPV of 4.0% (above the normative 3% NICE referral threshold) [3,20]. In contrast, a breast lump (FBC use 3%) in 40+-year-olds has a PPV of 4.8% [21]. 

The use of nongeneric tests was greatest for symptoms that represent expected features of specific cancers. PSA was used in far greater proportions (88%) in men presenting with the LUTS, which mirrors guideline recommendations [3]. However, two-thirds of men presenting with haematuria did not have PSA testing, which may represent opportunities for greater use consistent with guidelines that recommend GPs to consider PSA testing (alongside a digital rectal examination) to assess for prostate cancer in people with visible haematuria [22,23]. Relatively large proportions of patients subsequently diagnosed with cancer who presented with back pain were tested by serum protein electrophoresis or bone profile, possibly reflecting suspicion of multiple myeloma, given its cardinal presenting symptom is musculoskeletal pain.

Exploring blood test use by symptom PPV group highlights a positive association between blood test use and symptom PPV for ranges between 0.01 and 2.99%, where testing occurs in over half (51–52%) of patients presenting with symptom PPVs ranging 0.01–1.99% and increases to nearly two-thirds (64%) when symptom PPVs approach clinical thresholds for referral (i.e., 2.00–2.99%). In patients presenting with symptom PPVs ≥ 3%, test use declines as symptom PPVs increase (3.00–4.99% PPV = 42%, ≥5% PPV = 31%). Clinicians seem to prefer using tests in lower symptom PPV ranges, particularly just below the 3% threshold (i.e., 2.00–2.99%) where the relative information gain to assess the risk of cancer seems to be deemed greatest (i.e., symptoms do not meet referral criteria with regard to their PPV but are close to referral threshold). It may seem that blood tests are deemed less informative for patients already considered at higher risk if presenting with alarm symptoms. Therefore, the findings highlight opportunities for greater use in patients with non-alarm (nonspecific and vague) symptoms, belonging to the <3% PPV groups.

The potential for GP referral decisions to be guided by the combined diagnostic utility of blood test results and symptom information requires current phlebotomy services to be optimised (such as through increasing capacity for on-site phlebotomy within general practices). Practical modifications to the testing process can help simplify access to primary care investigations such as blood tests [24,25]. Future research can assess the cost-effectiveness of greater than the current use of blood tests in patients in different risk categories. 

## 5. Conclusions

Overall, the findings help to benchmark GPs’ use of blood tests in cancer patients. In the future, the proportions of patients with certain cancer sites who were tested with generic or organ-specific blood tests could help provide pilot targets for markers of diagnostic process quality. Generic blood tests may be underused in some populations (i.e., those presenting with symptoms with positive predictive value for cancer below the current (3%) referral thresholds). Given the growing evidence base supporting the use of blood tests to help assess cancer risk in symptomatic patients, research efforts to monitor blood test use over time and potential between-GP variation will be important for identifying opportunities to improve the diagnostic process. 

## Figures and Tables

**Figure 1 cancers-15-03587-f001:**
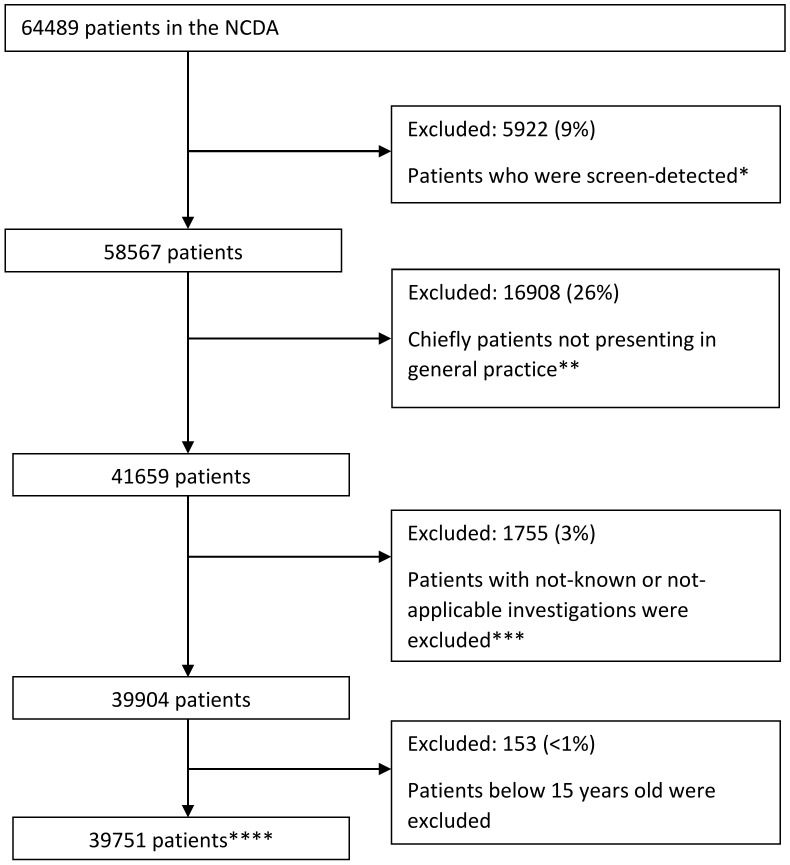
Derivation of the analysis sample (*n* = 39,751). * Screening detection status was assigned using a binary variable that categorised screen detection as “yes” or “no or unknown”. ** Includes a small number of cases with a recorded symptom that did not match the patient’s gender. *** Given the emphasis on describing associations between presenting symptoms and blood test use, patients with not known (N/K) or not applicable (N/A) investigations were excluded from subsequent analysis. **** Includes 571 patients with more than one tumour.

**Figure 2 cancers-15-03587-f002:**
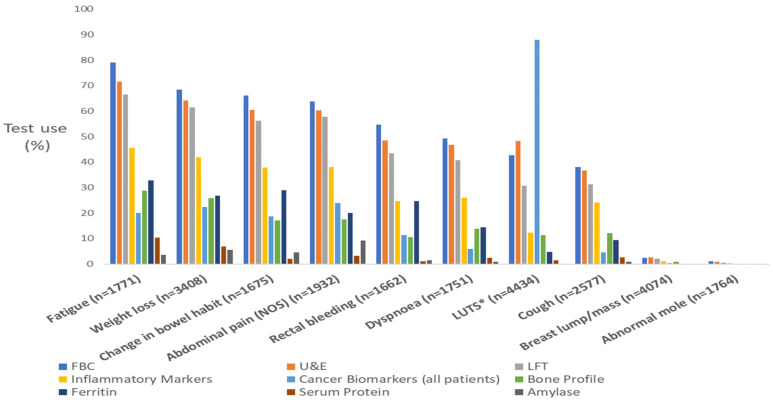
Proportion of patients having each of the 9 blood tests studied after presenting with the 10 most frequently reported symptoms. * Lower urinary tract symptoms; cancer biomarker use by LUTs is restricted to men (88%). Symptoms presented in descending order by FBC use.

**Table 1 cancers-15-03587-t001:** (**a**): Frequency of FBC, U&E, LFT, and IM blood tests based on presenting symptoms (proportions ranked by FBC order). (**b**): Frequency of bone profile, ferritin, serum protein, and amylase blood tests based on presenting symptoms (proportions ranked by FBC order).

(a)								
	Generic Blood Tests							
Symptom Name	FBC	%	U&E	%	LFT	%	IM	%
Fatigue (n = 1771) ^b^	1398	79	1267	72	1177	66	808	46
Loss of appetite (n = 1264) ^a^	939	74	887	70	842	67	583	46
Weight loss (n = 3408) ^c^	2331	68	2183	64	2090	61	1428	42
Upper abdominal pain (n = 1192) ^b^	805	68	741	62	736	62	492	41
Diarrhoea (n = 1013) ^b^	673	66	621	61	574	57	404	40
Change in bowel habit (n = 1675) ^d^	1107	66	1013	60	941	56	633	38
Nausea and/or vomiting (n = 1067) ^b^	693	65	666	62	648	61	433	41
Abdominal pain (NOS) (n = 1932) ^c^	1233	64	1163	60	1114	58	736	38
Lower abdominal pain (n = 1060)	683	64	631	60	584	55	407	38
Constipation (n = 831) ^b^	527	63	484	58	448	54	297	36
Distension (n = 980) ^c^	609	62	593	61	539	55	352	36
Back pain (n = 1405) ^a^	837	60	797	57	719	51	578	41
Dyspepsia (n = 805) ^b^	468	58	436	54	414	51	247	31
Rectal bleeding (n = 1662) ^e^	907	55	805	48	723	44	411	25
Bone pain (n = 490) ^a^	262	53	249	51	222	45	185	38
Other symptom (n = 2131)	1093	51	1021	48	926	43	589	28
Dyspnoea (n = 1751) ^b^	863	49	818	47	714	41	457	26
Dysuria (n = 551) ^b^	255	46	263	48	199	36	110	20
Urinary tract infection (n = 477) ^a^	209	44	217	45	148	31	93	19
LUTS (nocturia, frequency, hesitancy, urgency, retention) (n = 4434) ^e^	1893	43	2139	48	1360	31	545	12
Haematuria (n = 1465) ^d^	599	41	588	40	367	25	186	13
Neck lump/mass (n = 1201)	498	41	422	35	367	31	325	27
Not applicable (n = 2795)	1129	40	912	33	734	26	326	12
Chest pain (n = 960) ^b^	387	40	365	38	330	34	236	25
Not known (n = 687)	270	39	219	32	200	29	76	11
Dysphagia (n = 997) ^e^	383	39	366	37	341	35	190	19
Cough (n = 2577) ^b^	982	38	944	37	807	31	620	24
All other symptoms (n = 6828)	2310	37	2115	34	1900	30	1316	21
Chest infection (n = 686) ^a^	251	37	239	35	212	31	159	23
Other vaginal bleeding (n = 421)	131	31	91	22	74	18	37	9
Haemoptysis (n = 469) ^e^	143	30	131	28	111	24	80	17
Sore throat (n = 478)	132	28	113	24	96	20	88	18
Hoarseness (n = 468) ^c^	109	23	102	22	97	21	62	13
Post-menopausal bleeding (n = 896) ^e^	166	19	145	16	124	14	61	7
Breast pain (n = 768) ^b^	21	3	24	3	20	3	9	1
Breast lump/mass (n = 4074) ^e^	102	3	109	3	80	2	44	1
Abnormal mole (n = 1764)	18	1	17	1	8	0	5	0
All patients (n = 39,751)	15,540	39	14,555	37	12,414	31	7598	19
(**b**)								
	**Less Common Blood Tests**							
**Symptom Name**	**Bone Profile**	**%**	**Ferritin**	**%**	**Serum Protein Electrophoresis ***	**%**	**Amylase**	**%**
Fatigue (n = 1771) ^b^	509	29	581	33	185	10	62	4
Loss of appetite (n = 1264) ^a^	367	29	379	30	92	7	88	7
Weight loss (n = 3408) ^c^	878	26	913	27	236	7	187	5
Upper abdominal pain (n = 1192) ^b^	210	18	252	21	44	4	172	14
Diarrhoea (n = 1013) ^b^	162	16	279	28	21	2	64	6
Change in bowel habit (n = 1675) ^d^	287	17	483	29	35	2	75	4
Nausea and/or vomiting (n = 1067) ^b^	229	21	238	22	37	3	97	9
Abdominal pain (NOS) (n = 1932) ^c^	339	18	387	20	61	3	179	9
Lower abdominal pain (n = 1060)	195	18	233	22	23	2	39	4
Constipation (n = 831) ^b^	185	22	204	25	34	4	39	5
Distension (n = 980) ^c^	169	17	179	18	17	2	55	6
Back pain (n = 1405) ^a^	425	30	207	15	251	18	37	3
Dyspepsia (n = 805) ^b^	120	15	175	22	22	3	59	7
Rectal bleeding (n = 1662) ^e^	175	11	411	25	18	1	24	1
Bone pain (n = 490) ^a^	155	32	60	12	79	16	10	2
Other symptom (n = 2131)	368	17	368	17	136	6	56	3
Dyspnoea (n = 1751) ^b^	243	14	254	15	44	3	14	1
Dysuria (n = 551) ^b^	60	11	33	6	11	2	11	2
Urinary tract infection (n = 477) ^a^	42	9	33	7	11	2	5	1
LUTS ** (nocturia, frequency, hesitancy, urgency, retention) (n = 4434) ^e^	498	11	213	5	69	2	18	<1%
Haematuria (n = 1465) ^d^	118	8	79	5	11	1	9	1
Neck lump/mass (n = 1201)	125	10	78	6	30	2	5	<1%
Not applicable (n = 2795)	259	9	334	12	151	5	20	1
Chest pain (n = 960) ^b^	138	14	96	10	49	5	19	2
Not known (n = 687)	49	7	69	10	29	4	2	<1%
Dysphagia (n = 997) ^e^	98	10	136	14	5	1	22	2
Cough (n = 2577) ^b^	312	12	240	9	68	3	25	1
All other symptoms *** (n = 6828)	722	11	551	9	233	4	153	2
Chest infection (n = 686) ^a^	74	11	48	7	15	2	8	1
Other vaginal bleeding (n = 421)	20	5	46	11	5	1	1	<1%
Haemoptysis (n = 469) ^e^	38	8	29	6	3	1	3	1
Sore throat (n = 478)	27	6	28	6	6	1	3	1
Hoarseness (n = 468) ^c^	26	6	30	6	5	1	2	<1%
Post-menopausal bleeding (n = 896) ^e^	34	4	44	5	2	<1%	1	<1%
Breast pain (n = 768) ^b^	4	1	7	1	3	<1%	0	0
Breast lump/mass (n = 4074) ^e^	35	1	17	<1%	6	<1%	2	<1%
Abnormal mole (n = 1764)	4	<1%	4	<1%	0	0	0	0
All patients (n = 39,752)	4367	11	4299	11	1240	3	761	2

The blue–white–red boundaries are set at the upper, median, and lower values for each blood test. All other values are coloured proportionally. * Serum protein electrophoresis is coded as “Serum protein/paraprotein” in the NCDA. ** LUTS = lower urinary tract symptoms. *** Forty-six symptoms accounting for less than 1% (n = 398) of cases were grouped together, including nipple changes ^d^, pelvic pain, lymphadenopathy (localised) ^a^, jaundice ^e^, night sweats, gastroesophageal reflux ^a^, testicular lump, headache ^a^, erectile dysfunction, prog/sub-acute loss of central neuro function, pruritis, lip/oral cavity/tongue lump/mass, testicular pain, loin pain, fever ^a^, non-pigmented lesion ^e^, axillary lump/mass, unexplained lump suspicious of sarcoma, lesions suspicious of BCC, nipple discharge ^e^, thyroid lump/mass, lip/oral cavity/tongue ulcer, ulceration, early satiety, bruising, bleeding or petechiae, pallor, vaginal discharge ^b^, anal mass, deep vein thrombosis ^a^, visual disturbance or loss ^a^, vulval mass, epistaxis, vulval ulceration, penile ulceration, vaginal mass, lymphadenopathy (generalised) ^a^, fit/seizure ^b^, new onset diabetes, stridor, leukoplakia ^a^, fracture ^a^, lymph node pain with alcohol, renal colic, clubbing, haematemesis ^c^ and vulval bleeding. a: denotes a PPV for cancer of 0.01–0.99%; b: denotes a PPV for cancer of 1.00–1.99%; c: denotes a PPV for cancer of 2.00–2.99%, d: denotes a PPV for cancer of 3.00–4.99%, e: denotes a PPV for cancer ≥5%, based on prior research [18] and clinicalguidelines [3] (for Appendix A).

**Table 2 cancers-15-03587-t002:** Frequency of cancer biomarker use based on symptom presentation (proportions ranked by cancer biomarker order).

Symptom Name	Cancer Biomarkers					
	Men			Women		
	Number of Men with Symptom	Biomarker Use	%	Number of Women with Symptom	Biomarker Use	%
Fatigue (n = 1771) ^b^	939	249	27	832	105	13
Loss of appetite (n = 1264) ^a^	673	155	23	591	136	23
Weight loss (n = 3408) ^c^	2042	500	24	1366	259	19
Upper abdominal pain (n = 1192) ^b^	619	80	13	573	105	18
Diarrhoea (n = 1013) ^b^	563	92	16	450	74	16
Nausea and/or vomiting (n = 1067) ^b^	461	44	10	606	110	18
Change in bowel habit (n = 1675) ^d^	939	149	16	736	163	22
Abdominal pain (NOS) (n = 1932) ^c^	936	159	17	996	302	30
Lower abdominal pain (n = 1060)	438	132	30	622	199	32
Constipation (n = 831) ^b^	443	101	23	388	102	26
Distension (n = 980) ^c^	351	53	15	629	332	53
Back pain (n = 1405) ^a^	851	404	47	554	64	12
Dyspepsia (n = 805) ^b^	475	43	9	330	45	14
Bone pain (n = 490) ^a^	305	108	35	185	13	7
Rectal bleeding (n = 1662) ^e^	954	118	12	708	69	10
Other symptom (n = 2131)	1232	353	29	899	81	9
LUTS * (nocturia, frequency, hesitancy, urgency, retention) (n = 4434) ^e^	4242	3716	88	192	50	26
Dyspnoea (n = 1751) ^b^	973	73	8	778	30	4
Dysuria (n = 551) ^b^	408	261	64	143	19	13
Urinary tract infection (n = 477) ^a^	293	186	63	184	26	14
N/A (n = 2795)	2197	1231	56	598	33	6
Haematuria (n = 1465) ^d^	1156	410	35	309	8	3
N/K (n = 687)	532	336	63	155	7	5
Neck lump/mass (n = 1201)	680	25	4	521	8	2
Chest pain (n = 960) ^b^	536	47	9	424	19	4
Cough (n = 2577) ^b^	1441	86	6	1136	29	3
Dysphagia (n = 997) ^e^	670	29	4	307	16	5
All other symptoms (n = 6828) **	3332	715	21	2949	239	8
Chest infection (n = 686) ^a^	374	23	6	312	9	3
Other vaginal bleeding (n = 421)	0	** ≤3	≤0	421	56	13
Haemoptysis (n = 469) ^e^	302	7	2	167	≤3	1
Sore throat (n = 478)	328	≤3	≤1	150	4	3
Hoarseness (n = 468) ^c^	351	9	3	117	≤3	2
Post-menopausal bleeding (n = 896) ^e^	0	≤3	≤0	896	76	8
Breast pain (n = 768) ^b^	11	≤3	≤27	757	≤3	<1
Breast lump/mass (n = 4074) ^e^	52	≤3	≤6	4022	12	<1
Abnormal mole (n = 1764)	811	≤3	≤1	953	0	0
All patients (n = 39,752)	21,854	7828	36	17,898	1461	8

The blue–white–red boundaries are set at the upper, median, and lower values for each blood test. All other values are coloured proportionally. * LUTS = lower urinary tract symptoms. ** Biomarker use between 0–3 is represented as “≤3” to reduce the risk of residual disclosure, with corresponding proportions being calculated based on the use of 3 (or fewer) biomarker tests. a: denotes a PPV for cancer of 0.01–0.99%; b: denotes a PPV for cancer of 1.00–1.99%; c: denotes a PPV for cancer of 2.00–2.99%, d: denotes a PPV for cancer of 3.00–4.99%, e: denotes a PPV for cancer ≥5%, based on prior research [18] and clinical guidelines [3] (for Appendix A).

**Table 3 cancers-15-03587-t003:** Proportions and crude/adjusted ORs examining variation in blood test use by symptom PPV group.

	Population Total, n (Column %)	Received a Blood Test *, n (Row %)	Crude OR (95% CI) and *p*-Value	Adjusted OR *** (Excluding Cancer Site) and *p*-Value	Adjusted OR (Including Cancer Site) and *p*-Value
**Total:**	29,043 (100%) **	12,998 (45%)			
Symptom PPV *			**** <0.001	**** <0.001	**** <0.001
0.01–0.99%	3162 (11%)	1656 (52%)	2.41 (2.22–2.61)	2.40 (2.21–2.60)	1.57 (1.43–1.73)
1.00–1.99%	7454 (26%)	3779 (51%)	2.25 (2.12–2.39)	2.33 (2.19–2.48)	1.96 (1.81–2.12)
2.00–2.99%	4846 (17%)	3113 (64%)	3.94 (3.67–4.23)	3.89 (3.61–4.18)	2.79 (2.55–3.05)
3.00–4.99%	2101 (7%)	887 (42%)	1.60 (1.45–1.76)	1.47 (1.34–1.62)	1.25 (1.11–1.40)
≥5.00%	11,480 (40%)	3593 (31%)	ref	ref	ref

* Blood test variables included FBC, U&E, LFT, IM, bone profile, serum protein electrophoresis, ferritin, or amylase tests. ** A total of 10,709 cases with symptoms not captured in the PPV groups (i.e., because no published evidence supported symptom allocation) were excluded from the Appendix A. *** Adjustments are made for age, sex, IMD, symptom PPV, and cancer site (when specified). **** Post estimations using Wald tests explained the significance of the symptom PPV group in predicting blood test use. See Table 1b for symptom allocation to PPV categories. CI = confidence interval. IMD=Index of Multiple Deprivation. OR = odds ratio. Ref = reference group.

## Data Availability

Data are available through the Data Access Request Service of NHS England.

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
