# Peer review of "Pre-Referral Primary Care Blood Tests and Symptom Presentation before Cancer Diagnosis: National Cancer Diagnosis Audit Data"

_cancers, 2023, doi:10.3390/cancers15143587_

Round 1
Reviewer 1 Report (Previous Reviewer 1)
Thank you for addressing the comments and revising the manuscript. The quality is much improved. I don't have any further comments.
Reviewer 2 Report (Previous Reviewer 2)
Thank you the updated version. I feel you were able to address all the issues.
This manuscript is a resubmission of an earlier submission. The following is a list of the peer review reports and author responses from that submission.
Round 1
Reviewer 1 Report
Thank you for the opportunity to review “Primary care blood tests and symptom presentation before cancer diagnosis: National Cancer Diagnosis Audit data”. This manuscript proposed a description of 83 presenting symptoms on the frequency of four generic and five organ-specific blood tests utilizing National Cancer Diagnosis Audit data on 39,751 patients with incident cancer in 2018. The authors concluded that the use of generic blood tests can reflect a positive predictive value of presenting symptoms, and the use of organ-specific tests is more effective in patients with symptoms that are known to be associated with certain types of cancer. The topic itself is interesting, however my enthusiasm on it is dampened by a few major concerns. This manuscript has large room to be improved. Below, I discuss a few major concerns, followed by a few minor points about the paper.
Major concerns:
1. The introduction of the manuscript does not adequately state the clinical relevance of the study. The authors should provide more information on the rationale or significance/ innovation of the study.
2. The manuscript lacks a presentation of the characteristics of the 39,751 cancer patients, e.g., demographics, health condition, and cancer type. The authors should provide this information as it may reveal a great heterogeneity in the association of blood test use and cancer type, and this might affect the results of the study.
3. The study is titled “Primary care blood tests and symptom presentation before cancer diagnosis: National Cancer Diagnosis Audit data”, however there is no description about how the order of blood test and cancer diagnosis was determined.
4. Instead of solely relying on descriptive numbers, the authors could perform statistical tests to make the comparison, which would provide a more robust evidence for drawing conclusions.
5. The data is collected by participating GPs based on information in primary care records. The findings of a study may only be generalizable to certain populations in England. In addition, it lacks the explanation about those patients with not-known or not-applicable investigation.
Minor concerns:
1. The frequency of four generic blood tests based on presenting symptom appear cluttered and confusing for the readers. The color scheme does not accurately depict the relative ranking of proportions.
2. I would suggest that the 83 pre-defined symptoms should be grouped into multiple categories to create output that can be easily understood.
3. The author should provide the rationale for the inclusion criteria of including patients over 15 years old but below 18 years in the study. Those patter of healthcare encounter for these adolescents may differ significantly from older adults.
Reviewer 2 Report
Hello Authors,
Thank you sharing this article which shows blood test result and symptoms before cancer diagnosis. After reading I feel you are in right direction but if you expand your scope of the study you will find more evidence related to biomarker or symptoms that can help detect cancer from blood test. This could also create even more clear opportunities to generate better guidelines.
Reviewer 3 Report
This article starts from apparently banal, but to me striking fact, that, although the evidence would support the predictive value of some cancer tests in subjects presenting to their GP with generic symptoms, the prescription of these tests is limited to those patients that present with more "alarm" and specific symptoms. Therefore the generic symptoms patients that are eventually diagnosed with cancer face a longer path to diagnosis and treatment and lose potential benefits of aforementioned cancer tests if applied timely. The fact that we consider here cancer detected out of screening programs increases the value of the issue raised
Design is very clear and convincing, the studied population relevant in size and composition, grouping of symptoms is well described and rational.
Too bad that the info on time between relevant symptom presentation and blood test blood tests prescription/execution and a referral decision is missing.
I very much liked the conclusion" that commonly used generic blood tests may be underused in some populations (i.e., patients presenting with symptoms of lower specificity) and overused in others (in patients presenting with alarm symptoms). Indeed, I would have hoped that authors have gone further in elaboration of the study results and do some "modelling"
Possible outcome of such a study could be tentative pooling of the symptoms that , when observed together, have a sufficient ppv for cancer in order to justify the prescription of a more specific and more informative test. on the other hand it would be excellent if you could "calculate" on how many cancer patients did not benefit from such tests in primary care only because their symptoms were judged unspecific?
Some health economics comments on how optimisation of use of primary care diagnostic tests (and matching to symptoms) would improve on cancer detection rate in non screening settings, and therefore also improve the access of patients to timely care. I think this is very valuable issue and way to improve the outcomes of cancer patients using what we already have in clinics without waiting for brand new tests and screening modalities.
I think some additional calculations and an extra paragraph in discussion would make the authors effort even more relevant!